# Myristoyl's dual role in allosterically regulating and localizing Abl kinase

**Svenja de Buhr[1], Frauke Gräter[1,2]\***

[1]Heidelberg Institute for Theoretical Studies (HITS), Heidelberg University, Heidelberg, Germany; [2]Institute for Scientific Computing (IWR), Heidelberg University, Heidelberg, Germany

**Abstract** c-Abl kinase, a key signaling hub in many biological processes ranging from cell development to proliferation, is tightly regulated by two inhibitory Src homology domains. An N-terminal myristoyl modification can bind to a hydrophobic pocket in the kinase C-lobe, which stabilizes the autoinhibitory assembly. Activation is triggered by myristoyl release. We used molecular dynamics simulations to show how both myristoyl and the Src homology domains are required to impose the full inhibitory effect on the kinase domain and reveal the allosteric transmission pathway at residue-level resolution. Importantly, we find myristoyl insertion into a membrane to thermodynamically compete with binding to c-Abl. Myristoyl thus not only localizes the protein to the cellular membrane, but membrane attachment at the same time enhances activation of c-Abl by stabilizing its preactivated state. Our data put forward a model in which lipidation tightly couples kinase localization and regulation, a scheme that currently appears to be unique for this non-receptor tyrosine kinase.

## Editor's evaluation

This is an important study of the mechanism of how binding of the fatty acid myristic acid (MYR) inhibits the activity of the kinase c-Abl, a critical regulator of many cellular processes. While the general aspects of this regulation are known from structure determination and biochemical studies, the exact molecular mechanism and the nature of the allosteric inhibition were not known. The authors use MD simulation to close this gap and provide a compelling mechanistic description of the inhibitory mechanism.

**\*For correspondence:**
Frauke.Graeter@h-its.org

**Competing interest:** The authors declare that no competing interests exist.

## Introduction

The non-receptor tyrosine kinase c-Abl, hereafter referred to as Abl, is involved in a plethora of signaling processes, including cell proliferation and survival, stress response, neuronal development, and remodeling of the actin cytoskeleton (*Khatri et al., 2016*; *Pendergast, 2002*; *Wang, 2014*; *Woodring et al., 2003*). Knock-out of Abl leads to developmental lethality (*Koleske et al., 1998*), and fusion with the breakpoint cluster region gene from chromosome 22 yielding the BCR-Abl oncogene de-regulates kinase activity and is the main cause of chronic myelogenous leukemia (*Greuber et al., 2013*). Given its key signaling function, tight and multi-modal regulation of Abl kinase activity is essential, and allostery has emerged as an important regulatory principle and therapeutic target for this large multi-domain kinase (*Nussinov et al., 2022*).

From N- to C-terminus, Abl consists of a flexible linker referred to as the N-cap, two Src homology (SH) domains, SH3 and SH2, the catalytic kinase domain, a disordered region containing proline-rich motifs and localization signals, and an F-actin-binding domain (*Figure 1*). The assembled, autoinhibited structure is similar to Src family kinases and the two SH domains attach to the back of the kinase

**Figure 1.** Abl structure in the assembled state. The SH3 and SH2 domains (magenta) attach to the kinase domain (blue), mediated by two prolines (PxxP, purple) on the SH2-kinase linker and the kinked conformation of the $\alpha_I$ helix (cyan), respectively. Myr (yellow) binds to a hydrophobic pocket at the C-terminal of the kinase and the N-cap (black) clamps around the assembled complex. ATP (light gray) is bound at the active center between the kinase N- and C-lobe. Changes during kinase activation include detachment of the SH domains from the back of the kinase domain, unbinding of Myr and straightening of the $\alpha_I$ helix, rotation of the $\alpha_C$ helix (dark blue), assembly of the R-spine (green), extension of the A-loop (orange) to allow substrate binding and transition of the DFG-motif (orange stick representation) to the in-conformation. The C-terminal disordered region and F-actin binding domain (FABD) were not considered in our simulation systems.

domain with respect to the active site (*Nagar et al., 2003*). Isoform 1b of Abl is myristoylated at its N-terminus. The myristoyl moiety (Myr) binds to a hydrophobic pocket in the kinase domain, which induces a kink in the kinase C-terminal $\alpha_I$ helix. This kinked conformation of the helix enables docking of the SH2 domain to the kinase C-lobe and consequently kinase inhibition (*Hantschel et al., 2003*). Apart from this, two more characteristics contribute to stable inhibition. The PxxP motif on the linker connecting SH2 domain and kinase facilitates stable docking of the SH3 domain (*Panjarian et al., 2013b*; *Ren et al., 1993*), and the N-cap acts as a rigidifying clamp around the SH3-SH2-kinase complex (*Chen et al., 2008*). While many aspects of Abl activation have been examined, a comprehensive picture starting from this initial clamped and inhibited structure is missing.

Myr and the SH domains have to detach from the kinase domain for Abl activation. The kinase domain then undergoes several structural transitions: The $\alpha_C$ helix of the kinase N-lobe rotates inward, the activation (A-)loop switches from a collapsed to an extended state that allows substrate binding, the DFG-motif converts to the in-conformation capable of coordinating ATP and a $Mg^{2+}$ ion for catalysis, and the hydrophobic regulatory (R-)spine at the hinge between N- and C- lobe assembles (*Azam et al., 2008*; *Levinson et al., 2006*; *Taylor et al., 2019*; *Xie et al., 2020*). Docking of the SH2 domain to the kinase N-lobe facilitates stable transition to the active conformation (*Grebien et al., 2011*; *Lamontanara et al., 2014*; *Nagar et al., 2006*). Knowledge about these conformational states stems mainly from structural, that is, rigid, data and has been complemented with hydrogen-deuterium exchange mass spectrometry or solution NMR experiments investigating the effects of ATP-competitive or allosteric inhibitors (*Iacob et al., 2009*; *Iacob et al., 2011*; *Skora et al., 2013*) or determining the energy landscape of different Abl constructs (*Saleh et al., 2017*; *Xie et al., 2020*).

Previous computational studies have been instrumental in examining the conformational transitions between active and inactive states and structural elements regulating kinase activity such as $\alpha_C$ helix or DFG-motif conformation (*Dölker et al., 2014*; *Liu et al., 2022*; *Meng et al., 2018*; *Narayan et al., 2020*). However, surprisingly little is known about the early stages of Abl's allosteric activation pathway directly following unbinding of its natural ligand Myr. We here aim to resolve at atomistic level how Myr allosterically regulates Abl by impacting kinase domain dynamics and autoinhibition, and the conformation of the C-terminal $\alpha_I$ helix.

In contrast to the assembled state, in which the $\alpha_I$ helix assumes a kinked conformation, crystal structures of only the kinase domain without Myr or any other kink-inducing ligand bound display the helix in a straight conformation. This led to the common view that Myr unbinding causes straightening of the C-terminal $\alpha_I$ helix, impairing the SH2-kinase interface and consequently leading to the loss of inhibitory interactions from both SH domains. This is consistent with biochemical experiments showing elevated activity of Abl after Myr removal. However, a high-resolution structure of the kinase domain in complex with the SH domains, but without Myr has not been resolved. SAXS shape reconstructions and solution NMR of non-myristoylated Abl reveal the complex in the assembled state (*Badger et al., 2016*; *Skora et al., 2013*). We here show, using equilibrium and non-equilibrium molecular dynamics (MD) simulations, that Myr removal does not fully straighten the $\alpha_I$ helix but fosters a highly dynamic helix conformation, a preactivated kinase conformation, and a loosened interface to the SH domains. We also reveal the underlying allosteric pathways in atomistic detail.

Having established a key regulatory capacity of Myr, we additionally asked how Myr binding to Abl is controlled. Our MD simulations of Myr-unbinding from Abl versus a membrane suggest that Myr unbinding and kinase preactivation can be promoted by insertion of Myr into the cellular membrane, as Abl and a lipid bilayer can thermodynamically compete for Myr binding. Taken together, our computational study suggests a dual role of Myr as allosteric inhibitor and membrane anchor, putting forward the intriguing possibility of a direct crosstalk between Abl kinase activity and membrane localization.

## Results
### Effect of Myr on $\alpha_I$ helix and kinase dynamics

We first set out to explore the effect of Myr binding on the overall dynamics of the Abl kinase domain using equilibrium MD simulations. To this end, we simulated the SH3-SH2-kinase complex and systems of only the kinase with the $\alpha_I$ helix in a kinked or a straight conformation (*Figure 2A*). All models included ATP as the natural ligand of Abl and were simulated both with or without Myr. During the 40 µs cumulative simulation time (20 × 2 µs) per model, we did not observe spontaneous straightening of the $\alpha_I$ helix in the absence of Myr. The straight $\alpha_I$ helix, on the other hand, unfolds regardless of whether Myr is present or not (*Figure 2B*, *Figure 2—figure supplement 1*). This is not unexpected since the hydrophobic residues I521, V525, and L529 (we use 1b isoform numbering throughout this work), which interact with Myr in the kinked $\alpha_I$ conformation, are solvent exposed in the straight conformation. The low stability of a straight helix is consistent with crystal structures of the kinase domain, in which the folded or resolved part of the helix mostly ends before or around the position of the first hydrophobic residue, while the more C-terminal residues are not resolved. Longer helices are often stabilized by another protein copy packed against it in the crystal lattice and can therefore be regarded as a crystallization artifact (*Figure 2—figure supplement 2*). Importantly, the kinked $\alpha_I$ helix remains folded in the absence of Myr but is strongly destabilized, as can be seen by an increased root mean squared fluctuations (RMSF) of the second half of the helix after the kink both in simulations of a single kinase domain and even, albeit to a lesser extent, in the presence of the SH domains, which has also been observed by *Liu et al., 2022*.

The $\alpha_I$ helix as well as the Myr binding site are located at the C-terminus of the domain. To investigate whether changes on this site of the kinase affect the dynamics of the rest of the domain and thus its activity, we turned to principal component analysis. For comparison, we simulated the kinase additionally in an active conformation. This model is similar to the model with a straight $\alpha_I$ helix without Myr, but differs in the conformation of the DFG-motif and $\alpha_C$ helix, which are both in the 'in' conformation. Our analysis shows that the collective motions of the kinase along the first principal component (PC1), which describes an opening and closing of the N- and C-lobe with respect to each other, are closer to that of the active model when Myr was not bound (*Figure 2C and D*). When the

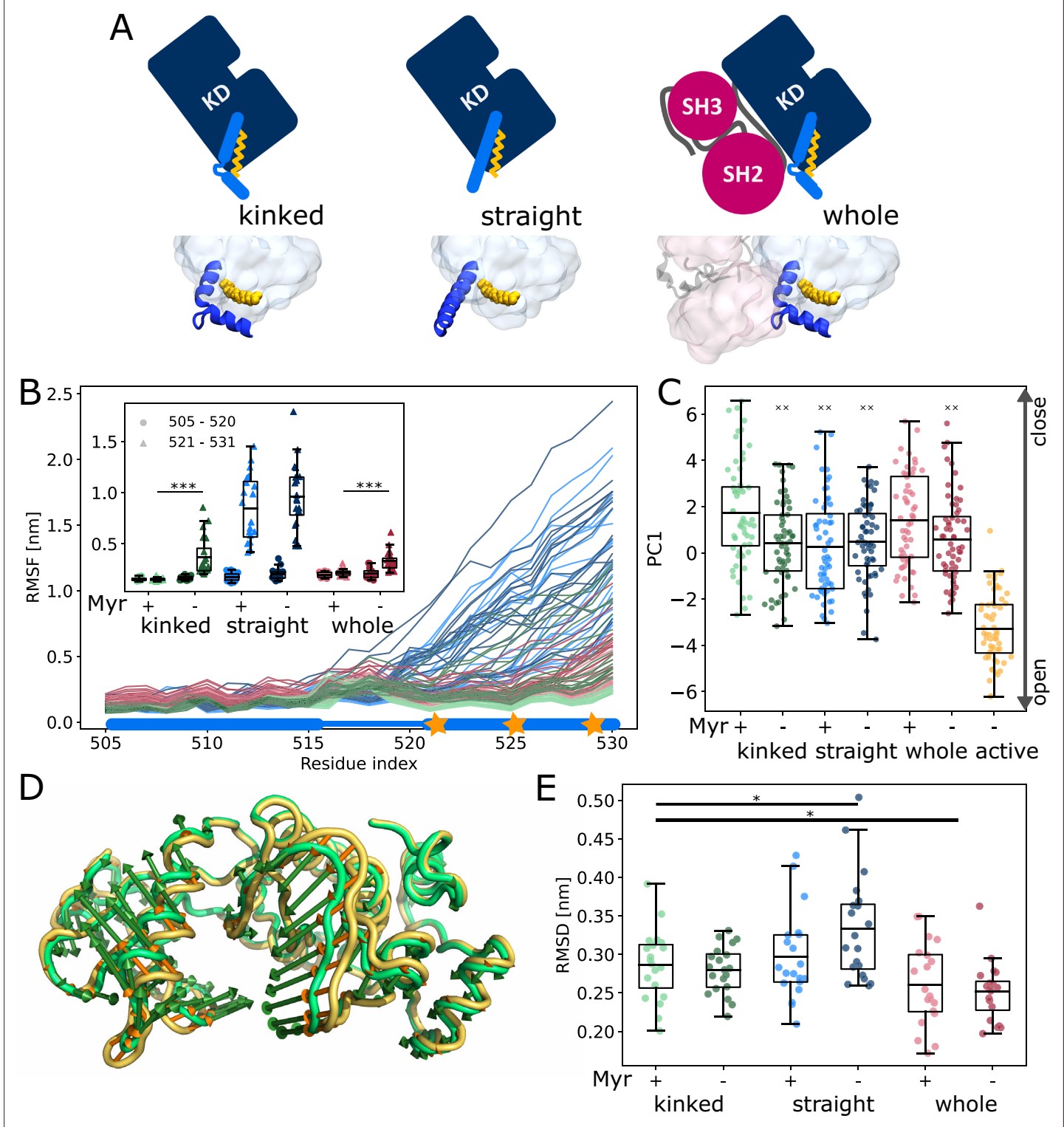

**Figure 2.** Influence of Myr on kinase domain dynamics. (**A**) Overview of simulated Abl models. Models referred to as 'whole' contain the kinase (KD, dark blue) and SH domains (magenta). The kinase was simulated with a kinked or straight $\alpha_I$ helix (light blue) conformation. All models were simulated in the absence or presence of Myr (yellow). Results of 'whole,' 'kinked,' and 'straight' models are shown in red, green, and blue colors, respectively, in subsequent figure panels. Lighter color shades represent simulations with and darker shades without Myr. (**B**) Root mean squared fluctuations (RMSF) of the $\alpha_I$ helix. The thick and thin blue line at the bottom indicates the position of the folded or unfolded part of the kinked helix, respectively. Orange stars represent the hydrophobic residues at the helix C-terminal part. The inset shows the average RMSF of residues of the first half (505–520, circles)

*Figure 2 continued on next page*

*Figure 2 continued*

and second half (521–531, triangles) of the helix. (**C**) Principal component analysis of kinase domain dynamics. The projection of the trajectories onto the first eigenvalue is shown. We separated the trajectory into 400 ns sections, leaving out 100 ns in between, and plotted the averages for each of these sections. The sections are not correlated according to an autocorrelation analysis (*Figure 2—figure supplement 3C*). (**D**) Porcupine plot illustrating the motion described by the first principal component. Orange tones represent the kinase domain in an active conformation, green tones represent the kinase with a kinked $\alpha_I$ helix bound with Myr. Visualized using PyMol v2.4.1 and the modevectors.py script (http://www.pymolwiki.org/index.php/Modevectors). (**E**) Root mean squared deviation (RMSD) of the kinase domain excluding the $\alpha_I$ helix. Centerlines of boxplots denote the mean, box edges the upper and lower quartile. Whiskers represent 1.5× interquartile range. */***p-value <0.05/0.001 between indicated models. ×× p-value vs. kinked + Myr < 0.01.

The online version of this article includes the following figure supplement(s) for figure 2:

**Figure supplement 1.** Helicity of the $\alpha_I$ helix.

**Figure supplement 2.** $\alpha_I$ helix lengths in crystal structures.

**Figure supplement 3.** Convergence of equilibrium molecular dynamics (MD) simulations.

**Figure supplement 4.** Comparison of data sets.

kinase domain is simulated with a straight $\alpha_I$ helix, this effect is observed even in the presence of Myr. Myr unbinding also leads to an increased root mean squared deviation (RMSD) of the kinase domain with a straight $\alpha_I$ helix, while a kinked helix seems to counteract this effect (*Figure 2E*). As expected, binding of the SH domains decreases the RMSD, clamping the kinase in a more rigid state.

We note that the differences between the tested conditions are subtle compared to the spread of values within each model. Yet, we observe a significant shift of the conformational ensemble of Abl toward the active state. Thus, Myr unbinding partially activates Abl kinase, while full activation requires the SH domains to detach.

## Force distribution analysis reveals collaborative effect of Myr and SH domains on kinase activity

To understand how Myr or the SH domains influence Abl dynamics and consequently activity, we used force distribution analysis (FDA) to decipher allosteric pathways (*Figure 3*). In short, we compute time-averaged forces between any pairs of residues as obtained from equilibrium MD simulations and determine the difference between the two conditions. Comparing simulations of the SH3-SH2-kinase complex with and without Myr shows high force differences spread over a large portion of the kinase C-lobe (*Figure 3A*). They are transmitted via the $\alpha_F$ helix and reach residue G402 of the DFG-motif as well as R381 and D382 of the catalytic triad. Further high force differences can be seen between the Myr binding site and the SH2 domain, which underlines previous reports that Myr can reinforce SH2 domain docking to the kinase C-lobe (*Nagar et al., 2003*). Notably, we also observe force differences distant from the Myr binding between the SH3 domain and kinase N-lobe as well as the hinge between N- and C-lobe. Performing the same Myr vs. no Myr comparison on simulations of a kinase domain without the SH domains, however, reveals that the allosteric transmission pathway to the active site is interrupted (*Figure 3B*). The force differences are mainly concentrated around the Myr binding site, involving all four helices making up the binding pocket. This reflects the increased flexibility of the second part of the $\alpha_I$ helix in the absence of Myr (compare *Figure 2B*). Overall, FDA results indicate that Myr binding by itself has a long-range allosteric effect on the kinase domain and its active site. This effect is in agreement with the observed changes in kinase domain dynamics and goes beyond locally controlling the conformation of the $\alpha_I$ helix. However, the SH domains, by locking the kinase domain, are required for the inhibitory effect of Myr to be properly transmitted to the active site. $\alpha_I$ helix straightening has a profound impact on the whole kinase C-lobe, regardless of whether Myr is present or not (*Figure 3—figure supplement 2*), explaining why helix straightening is sufficient to alter kinase dynamics (compare *Figure 2C*).

We then went on to focus on the allosteric impact of SH domain binding to the kinase both in the presence or absence of Myr. In simulations with Myr, medium force differences spread over the whole kinase domain (*Figure 3C*), including the hinge between the N- and C-lobe and the active site. They connect residues between the A-loop and $\alpha_C$ helix. In detail, the cluster links A-loop residue R405 to E305 and E311 from the $\alpha_C$ helix. This likely contributes to $\alpha_C$ helix outward rotation seen in inactive states of Abl by replacing the E305-K290 interaction stabilizing the inward rotation (*Levinson et al.,*

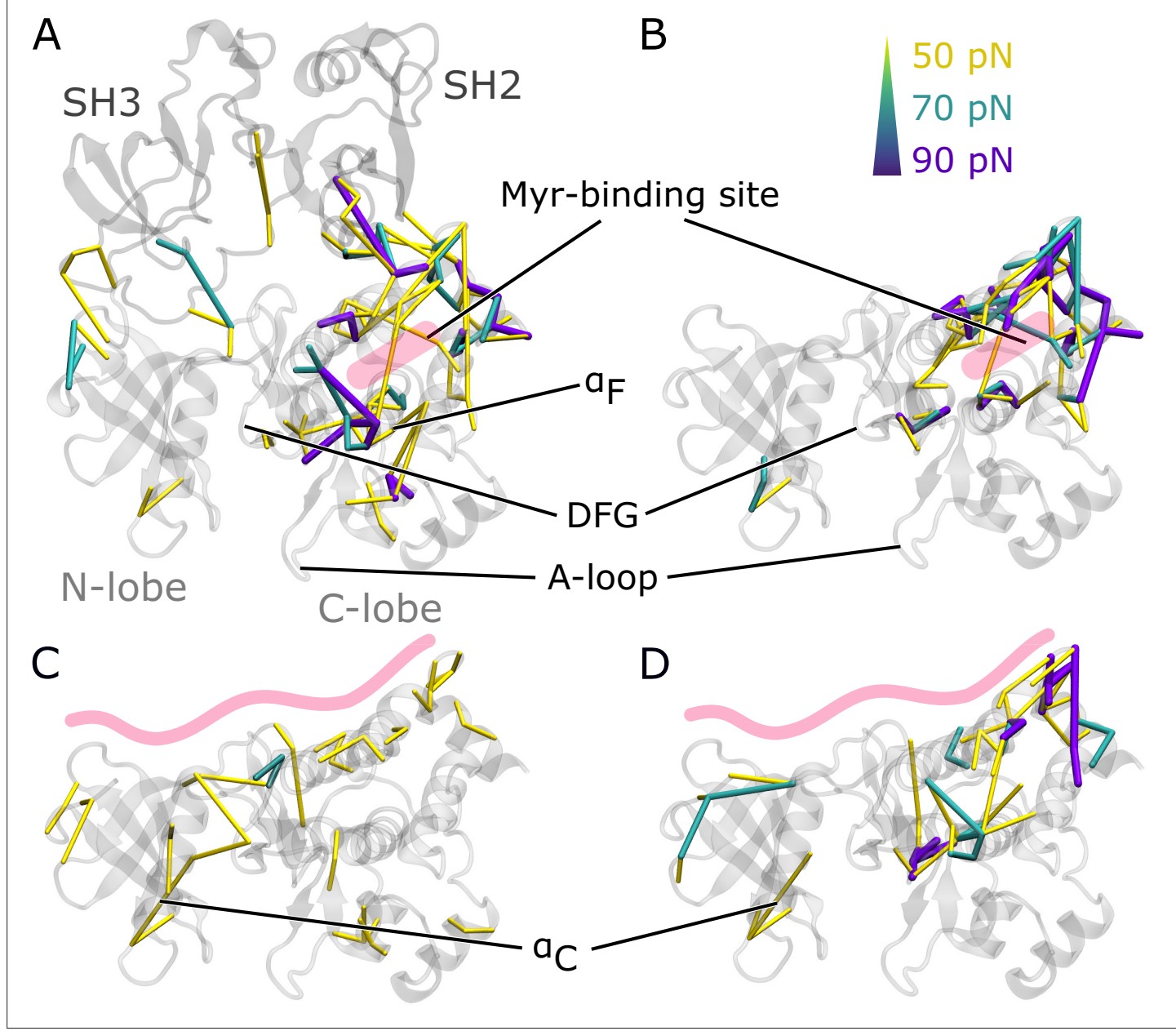

**Figure 3.** Residue pairwise force differences. The protein is shown in gray. Difference clusters with the indicated force threshold are visualized when at least three residues are connected. As indicated in light red, the top row compares differences between simulations with and without Myr in (**A**) the SH3-SH2-kinase complex or (**B**) a sole kinase domain with a kinked $\alpha_I$ helix. The bottom row compares simulations with and without the SH domains (**C**) with Myr or (**D**) without Myr.

The online version of this article includes the following figure supplement(s) for figure 3:

**Figure supplement 1.** Biggest force difference clusters with and without SH domains.

**Figure supplement 2.** Force differences between straight and kinked $\alpha_I$ helix.

2006; *Dölker et al., 2014*; *Liu et al., 2022*). In Abl's active conformation, R405 and pY412 form hydrogen bonds to stabilize the extended conformation of the A-loop, an interaction that is disrupted when R405 turns its sidechain toward E305 (*Panjarian et al., 2013a*; *Xie et al., 2020*). In the absence of Myr, we still observe force differences spread over the kinase-SH domain interface (*Figure 3D*). However, all force differences spanning the active site are gone. A similar picture is obtained when looking at the biggest connected force difference cluster instead of all clusters with a minimum size

(*Figure 3—figure supplement 1*). In the presence of Myr, we observe a large cluster spanning the kinase N- and C-lobe with the strongest interactions at the active site and hinge region. Without Myr, the cluster is retained at the C-lobe, indicating that Myr is needed in order for the SH domains to exert their clamp on the kinase domain. These results further underline that both Myr and the SH domains have an impact on the kinase domain that extends to the A-loop, but neither are sufficient to fully inhibit Abl by themselves and they instead exert a collaborative effect.

## $\alpha_I$ helix rearrangements following Myr unbinding can occur without SH domain detachment

It is commonly thought that Myr unbinding induces $\alpha_I$ helix straightening as the next step in the allosteric activation pathway of Abl (*Nagar et al., 2003*). This is supported by the fact that the straight helix is incompatible with kinase inhibition by SH2 domain binding to the C-lobe of the kinase as observed in crystal structures of Abl (*Figure 4A*). In contrast, as mentioned above, we did not observe spontaneous straightening of the $\alpha_I$ helix in the absence of Myr or the SH domains. However, the folding process might happen on a timescale longer than accessible with conventional MD simulations. To accelerate the process and ask how $\alpha_I$ helix straightening impacts SH2 binding, we turned to Metadynamics simulations. We enhanced the sampling of conformations with high helicity (*Pietrucci and Laio, 2009*) of residues 515–521, which correspond to the region interrupting the helical fold of the $\alpha_I$ helix in its kinked conformation.

We observed that straightening of the helix is possible in the presence of the SH domains, with transient straightening events occurring during the Metadynamics simulations without SH domain detachment (example shown in *Figure 4B*, left). To assess the stability of the straight conformations without the bias introduced by Metadynamics, we took 23 frames with high helicity across different trajectories and simulated them using equilibrium conditions for another 500 ns. Residues from M515 onward remained folded in 10 cases, (partially) unfolded in 9 cases, and fell back to a conformation resembling the kinked helix in 4 cases (*Figure 4B*, right). The kinase-SH2 interface is stable as quantified by the sum of contacts per frame (*Figure 4C and D*), and the SH2 domain has to only marginally adjust its position for helix straightening. This was already evident from only transient unbinding of the SH2 domain during a few Metadynamics simulations (*Figure 4—figure supplement 1*). When considering contacts that are formed in at least 10% of the simulation time, 61 out of 69 (88.4%) contact pairs are conserved compared to MD simulations of the SH3-SH2-kinase complex with a kinked $\alpha_I$ helix (*Figure 4E*). Unsurprisingly, most contact losses are observed between the SH2 domain and the $\alpha_I$ helix, which changed conformation during the Metadynamics simulation. The average number of native contacts per frame excluding the $\alpha_I$ helix ('core' in *Figure 4E*) reduces from 0.35 to 0.32, with a mean absolute error of 0.094. We conclude that helix conformational changes are certainly possible without prior unbinding of the SH2 domain.

## Myr can bind to the cellular membrane

Having established how Myr acts as an allosteric effector of Abl, the open question remains where the hydrophobic Myr is stored after detaching from its allosteric binding pocket in the kinase for preactivation. Conceivable possibilities are either other proteins or lipid membranes. To the best of our knowledge, no proteins acting as Myr secretion factors for Abl have been identified. Direct membrane binding of Abl is not widely recognized and the membrane-proximal fraction of Abl is not affected by Myr removal (*Hantschel et al., 2003*). However, enrichment of Abl at membranes has been observed upon displacement of Myr by an allosteric inhibitor (*Choi et al., 2009*). We therefore reasoned that Myr stays within its binding pocket due to the lack of other binding partners while Abl is localized in the cytoplasm. However, if Abl is recruited to membrane-proximal regions by binding to actin filaments with its F-actin binding domain upon cell adhesion (*Woodring et al., 2002*), Myr can be inserted into these membranes. We set out to explore the thermodynamic probability of this scenario by determining the free energy of Myr unbinding from either a lipid bilayer or its protein binding pocket using umbrella sampling simulations (*Figure 5*).

Displacing Myr from a simple 1-palmitoyl-2-oleoyl-glycero-3-phosphocholine (POPC) membrane involves a free energy of unbinding, or potential of mean force (PMF), of 36.3 ±0.5 kJ/mol, similar to the experimentally determined value of 33.5 kJ/mol (*Peitzsch and McLaughlin, 1993*). Assessing the free energy of Myr unbinding from Abl kinase by umbrella sampling for comparison is less straightforward

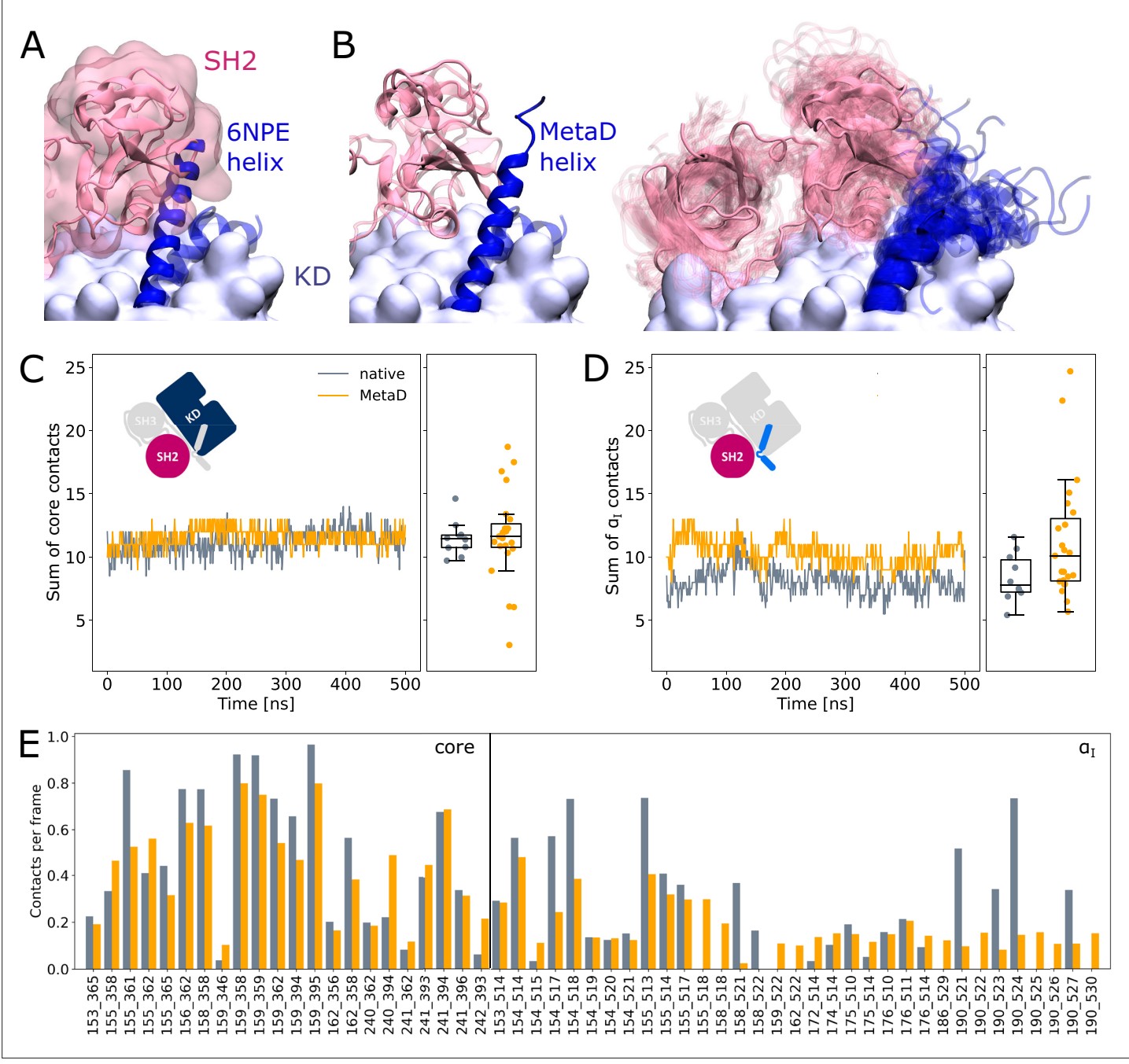

**Figure 4.** SH2 domain binding is preserved after Metadynamics enhanced $\alpha_I$ helix straightening. (**A**) The straight $\alpha_I$ helix from crystal structures (PDB-ID: 6NPE, blue) clashes with the Abl SH2 domain (light pink). For reference, the structure corresponding to a kinked helix is shown transparently. The kinase domain is shown in opaque surface representation. (**B**) Helix conformations obtained by Metadynamics simulations are compatible with SH2 binding. One example of a straightened helix is depicted on the left, the right image shows an overlay of all 23 frames with high helicity after simulating them under equilibrium conditions. (**C, D**) Sum of contacts per frame considering contacts of (**C**) the kinase domain excluding the $\alpha_I$ helix or (**D**) only the $\alpha_I$ helix. Contacts are considered native when they appeared in our simulations of the Abl SH3-SH2-kinase complex. The 'MetaD' contacts were determined from simulations started from Metadynamics frames with high helicity values. (**E**) Conservation of contacts per frame and residue pair. Contacts, which are present in at least 10% of the simulation time in either native simulations or after Metadynamics, are shown. The label 'core' denotes contacts excluding the $\alpha_I$ helix, which starts at residue index 505.

The online version of this article includes the following figure supplement(s) for figure 4:

**Figure supplement 1.** Distances between the kinase and SH2 domain.

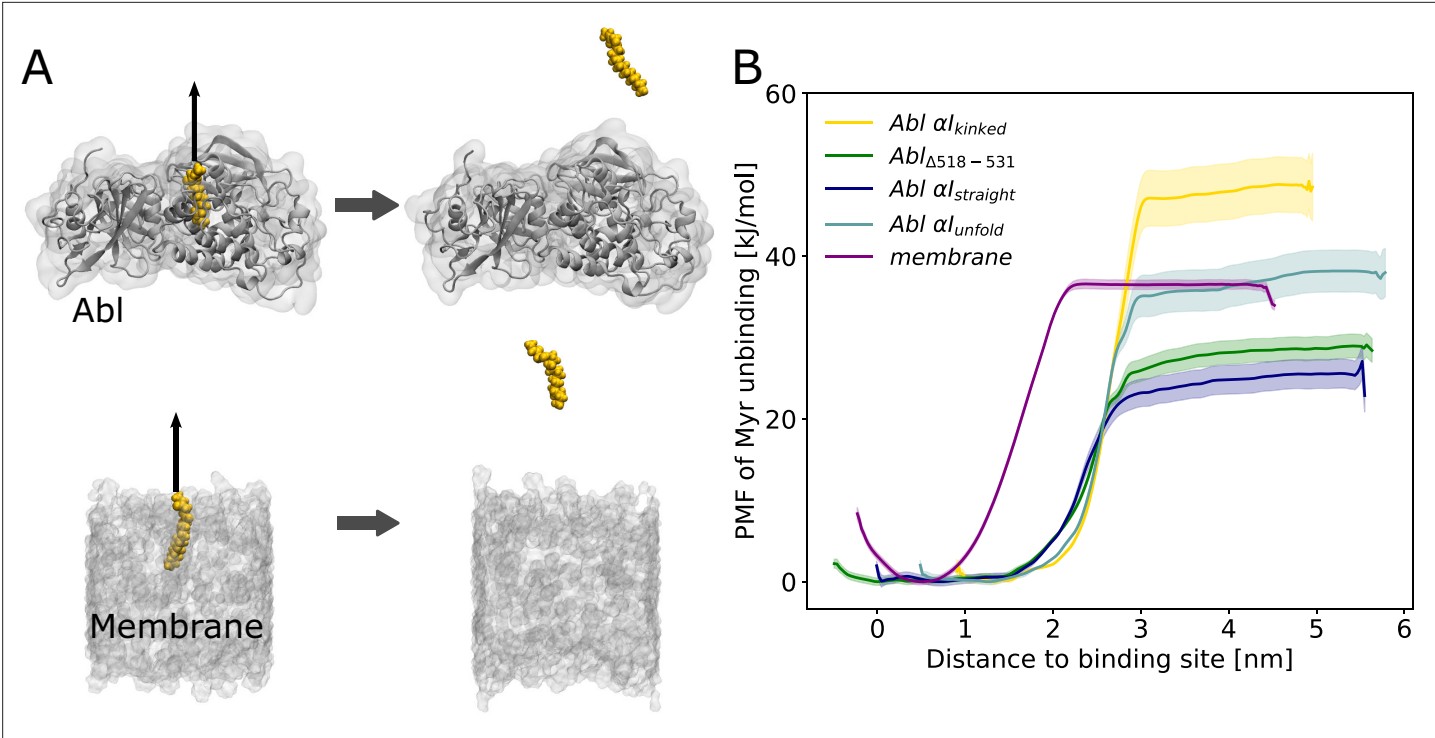

**Figure 5.** Myr binding to the membrane can compete with Myr binding to Abl. (**A**) Umbrella sampling was performed after pulling Myr from its protein binding pocket or a 1-palmitoyl-2-oleoyl-glycero-3-phosphocholine (POPC) membrane. (**B**) Potential of mean force (PMF) profiles of Myr unbinding. For comparison, we used an Abl SH3-SH2-kinase complex with a partially unfolded or straight $\alpha_I$ helix conformation as obtained by Metadynamics simulations, with the residues after the kink (518–531) deleted, or with the kinked conformation.

The online version of this article includes the following figure supplement(s) for figure 5:

**Figure supplement 1.** Convergence of umbrella sampling windows.

due to the involved conformational change of the $\alpha_I$ helix. In equilibrium MD simulations, we observed the kinked $\alpha_I$ helix conformation to be destabilized if Myr is absent (**Figure 2B**). X-ray structures with an empty Myr pocket support this and suggest helicity to prevail until residue ~519, while more C-terminal residues are largely disordered (**Figure 2—figure supplement 2**). To determine the free energies for Myr unbinding from Abl kinase that reflect this conformational propensity, we carried out umbrella sampling simulations using an Abl model with a partially unfolded $\alpha_I$ helix obtained during Metadynamics simulations. In addition, as extreme cases, a truncated version in which residues

**Table 1.** $\Delta G$ values derived from umbrella sampling simulations or observed experimentally (*__Peitzsch and McLaughlin, 1993__*, †__Hantschel et al., 2003__).

$K_D$ values are calculated using the Nernst equation with the assumption of an equal number of binding sites of 1 per Myr. Errors in the $\Delta G$ values were obtained by bootstrapping and propagated for $K_D$ calculation.

|  | $\Delta G$ (kJ/mol) | $K_D$ (µM) |
|---|---|---|
| Membrane | 36.3 ± 0.9 | 0.5 ± 0.03 |
| Membrane$_{Exp.}$ | 33.5* | 1.5 |
| Abl$_{Exp.}$ | 32.4 | 2.3† |
| Abl $\alpha_{I\text{ - kinked}}$ | 47.3 ± 3.7 | 6 ± 1 |
| Abl $_{\Delta 518 - 531}$ | 28.3 ± 1.9 | 11.9 ± 2.1 |
| Abl $\alpha_{I\text{ - straight}}$ | 24.9 ± 2.3 | 45.9 ± 11.3 |
| Abl $\alpha_{I\text{ - unfold}}$ | 37.8 ± 2.9 | 0.3 ± 5 |

518–531 are absent and a straightened helix conformation were used. For all of these models, the free energy of Myr unbinding from the protein is similar or smaller compared to unbinding from the membrane, supporting the notion that Myr binding to the membrane is thermodynamically possible. This result is also supported by previous measurements. The dissociation constant $K_D$ for binding of a myristoylated peptide from Abl's kinase domain has been determined to be 2.3 μM (*Hantschel et al., 2003*), corresponding to a binding-free energy of 32.4 kJ/mol (*Table 1*), in good agreement with the calculated free energies using Abl models with a truncated or partially unfolded $\alpha_I$ helix conformation. In contrast, the free energy for Myr unbinding from Abl was significantly higher when using an Abl model based on the crystal structure with Myr bound (i.e. with a kinked $\alpha_I$ helix). This can be explained by the hydrophobic residues lining the Myr binding pocket, which interact with Myr and are solvent exposed after unbinding.

Overall, our umbrella sampling results indicate that indeed helix rearrangements have to be taken into account to reflect the correct energy of unbinding. We note that these helix rearrangements of course contribute to the actual free energy of Myr unbinding but have not been covered during the limited timescale of umbrella sampling. The comparison to the experimental value, however, suggests that this approximation is feasible. The two experimental studies of Myr-Abl binding and Myr-membrane insertion furthermore confirm our major result that the involved free energies are comparable. Thus, we conclude that insertion of a single myristoyl moiety into a lipid bilayer can thermodynamically compete with binding of Myr into the known pocket of Abl's kinase domain.

## Discussion

In this study, we have used equilibrium and enhanced sampling simulation techniques to understand Abl's allosteric inhibition by Myr and the SH domains. We were not only able to describe the effect of different $\alpha_I$ helix conformations as well as Myr and SH domain binding on the overall dynamics of Abl's kinase domain, but could also explain the observations by identifying the pathways from the allosteric site to the active center at residue-level resolution. We observed that both Myr and the SH domains by themselves were able to impact residues at the A-loop. However, they act in concert for full inhibition: We saw how forces are transmitted from the Myr binding site to the active site along the $\alpha_F$ helix only in the presence of the SH domains. Vice versa, we observed how the SH domains act on the hinge between N- and C-lobe only when Myr was bound to the kinase domain, affecting known hallmarks of Abl kinase activity such as the $\alpha_C$ helix, the DFG-motif, and the catalytic triad (*Figure 3*). This is in line with and explains in molecular detail that a kinase domain bound to Myr or the allosteric inhibitors GNF-2 and GNF-5 without the SH domains or Abl mutants deficient in Myr binding display increased activity (*Choi et al., 2009*; *Fabbro et al., 2010*; *Hantschel et al., 2003*).

It has so far remained unclear how exactly Abl activation progresses upon Myr unbinding. Using equilibrium and Metadynamics simulations, we were able to show that the $\alpha_I$ helix gains flexibility upon Myr unbinding and that both straightening and partial unfolding of the $\alpha_I$ helix are possible even without prior unbinding of the SH domains. Both types of helix rearrangements are compatible with crystal structures of Abl's kinase domain (*Figure 2—figure supplement 2*). The helix fold continues longer compared to the kinked conformation seen in crystal structures of the Myr-bound SH3-SH2-kinase complex (*Nagar et al., 2003*), but is often not resolved far beyond the position of the kink, in agreement with the high flexibility of this most C-terminal part we observe in our simulations. The helix rearrangements upon Myr unbinding involve only a slight adaptation of the SH2 position as evidenced by shifts in residue pair contact counts (*Figure 4E*). This observation is in line with SAXS and solution NMR data showing that an Abl mutant incompetent in Myr binding or apo-Abl remains in the assembled conformation (*Badger et al., 2016*; *Skora et al., 2013*). We suggest that the different $\alpha_I$ helix conformations and/or binding poses of the SH2 domain explain why apo-Abl has failed to crystallize so far. The observation of an allosteric effect of Myr unbinding leading to a preactivated state with an altered SH-kinase interface implies that factors in addition to Myr unbinding contribute to full Abl activation. Transient unbinding events of the SH domains, as we also observed in a few Metadynamics simulations (*Figure 4—figure supplement 1B*), could enable phosphorylation of Y245 on the SH2-kinase linker (*Brasher and Van Etten, 2000*) or binding of activator substrates to block rebinding of the SH domains (*Wang, 2014*), thereby advancing Abl activation.

It is widely accepted that Myr unbinding leads to Abl activation. However, the non-myristoylated Abl isoform 1a, which only differs from the 1b isoform by the N-terminal half of the N-cap, is not

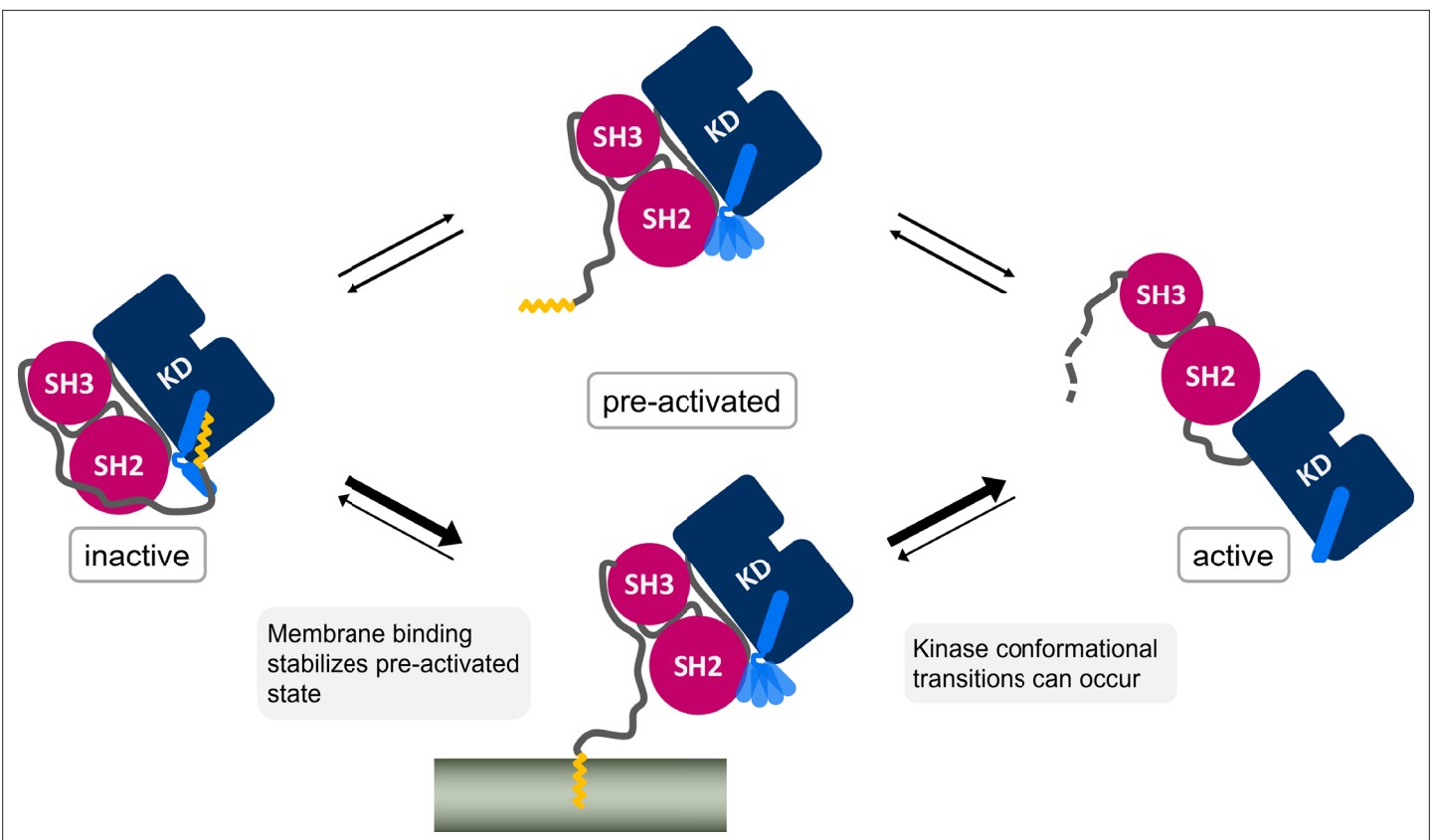

**Figure 6.** Myr, which is covalently bound to the N-terminal end of the flexible N-cap linker of Abl, leaves its protein binding pocket during the kinase activation pathway. Membrane binding of Myr enhances Abl activation by stabilizing the preactivated state.

de-regulated (*Van Etten, 2003*; *Nagar et al., 2003*). This suggests that additional factors contribute to Abl activation since the loss of Myr can apparently be compensated. We addressed the so far unanswered question where Myr binds to after it has left its protein binding pocket. The fatty acid has to be stored in a hydrophobic location or it will quickly rebind to its binding pocket and switch Abl back into its inactive state. We here propose the membrane as an anchoring point for Myr, identifying a new role for Abl localization and regulation (*Figure 6*), which is in line with the fact that protein myristoylation is usually involved in membrane recruitment (*Resh, 2016*). By harboring Myr, the membrane would stabilize the preactivated state, allowing time for activating phosphorylations of Y412 on the A-loop or Y245 (*Brasher and Van Etten, 2000*), permanent SH domain detachment, or SH2 domain binding to the kinase N-lobe. All of these events enhance kinase domain transitions to the fully active state. Our results show that membrane binding is energetically as favorable or potentially even more favorable than protein binding, a tendency also found when comparing experimental measurements (*Table 1*; *Hantschel et al., 2003*; *Peitzsch and McLaughlin, 1993*). The simulations allowed us to uniquely consider rearrangements of the $\alpha_I$ helix conformation, which strongly impacts this free energy of Myr unbinding, suggesting that reconfigurational free energies of the helix critically define the competition between kinase and membrane binding of Myr. Previous reports that Abl localizes to membranes upon Myr displacement from the protein by an allosteric inhibitor (*Choi et al., 2009*) or upon kinase domain deletion (*de Oliveira et al., 2013*) confirm that Abl is indeed capable of binding to membranes and that the membrane competes with the protein hydrophobic pocket for Myr binding. However, *Choi et al., 2009* revealed enrichment of Abl at both endoplasmic reticulum membranes and the outer cell membrane, indicating that Myr attached unspecifically to the hydrophobic environment of any membrane.

The high free energy difference we determined for Myr unbinding from the SH3-SH2-kinase model with a kinked $\alpha_I$ helix is consistent with Abl being well soluble in aqueous solution such as the cytosol (*Nagar et al., 2003*), which prevents unspecific binding. At the same time, it explains why

the membrane-proximal fraction of Abl is not decreased by Myr deletion (*Hantschel et al., 2003*). If Abl gets recruited to regions close to cellular membranes by binding to the cytoskeleton with its C-terminal F-actin binding domain, the free energy difference that has to be overcome reduces from ~58 kJ/mol to ~12 kJ/mol ($\Delta\Delta G$ between Myr unbinding from Abl $\alpha_{I\text{-}kinked}$ and membrane, *Table 1*). The difference is further balanced by helix rearrangements occurring simultaneously or subsequently to Myr unbinding from Abl. This agrees well with the fact that Abl gets activated after localization to focal contacts (*Lewis et al., 1996*; *Woodring et al., 2002*). Membrane binding could therefore be considered an extra regulatory layer, which can be carefully balanced by localization cues or membrane composition. In fact, N-cap residues 15–60, which have been shown to be irrelevant for Abl inhibition (*Hantschel et al., 2003*), encompass a number of basic residues (K24, K28, K29, R33) and could interact with acidic membrane lipids found at focal contacts. Src kinase also features basic residues near its myristoylation site, which stabilize the interaction with $PIP_2$-rich membranes (*Daday et al., 2022*). We speculate that a membrane also enhances Myr unbinding from Abl and Myr insertion by using these residues as a guide for Myr.

In summary, we have shown that Myr and the SH domains act in concert to inhibit Abl kinase activity and visualized the detailed allosteric transmission pathway from the Myr binding site to the active site. We propose a unique dual role of Myr for Abl: in addition to Abl inhibition, it also localizes Abl to the membrane. Proximal membranes accommodating Myr would thus aid the allosteric activation of Abl by stabilizing its preactivated state. This novel crosstalk can be directly tested by biochemical or cell experiments. For example, the transmission pathway from the Myr binding site toward the active site along the $\alpha_F$ helix could be disrupted through mutations. We specifically suggest residues S439, D440, and V441 from the N-terminus of the helix as these interact with residues from the catalytic triad. Furthermore, the kinase activity in the presence of lipid membranes could be determined and compared to that of non-myristoylated mutants or mutants in which the Myr binding site is blocked. We propose that the coupling between Abl activity and membrane anchoring is ideally suited to be exploited by new therapeutic approaches that allosterically target Abl.

## Methods

### Modeling of Abl structures for simulations

We based our Abl models on a crystal structure including the SH3, SH2, and kinase domains, as well as part of the N-Cap and Myr (PDB-ID: 2FO0). This structure has a D355N mutation, which we reversed using Modeller (*Sali and Blundell, 1993*). Furthermore, it is bound by the inhibitor PD-166326 (6-(2,6-dichlorophenyl)–2-[3-(hydroxymethyl)phenyl]amino-8-methylpyrido[2,3-D]pyrimidin-7(8H)-one). Since we aimed for a more physiological structure, we replaced this inhibitor by ATP by doing the following modifications: we overlayed the kinase domain with an ATP-analog-bound Abl structure (PDB-ID: 2G2F, RMSD of 1.7 Å across all atoms). We converted the ATP-analog 112 (thiophosphoric acid O-((adenosyl-phospho)phospho)-S-acetamidyl-diester) to ATP by deleting the acetamidyl group and changing the sulfur atom to oxygen. The A-loop of 2FO0 was pushed upward by the inhibitor and the ATP-positioning loop clashed with the ATP-phosphates. We therefore refined these two loops based on homology with the 2G2F structure. We modeled the structures with a straight $\alpha_I$ helix by creating a homology model of the modeled ATP-bound kinase and PDB structure 6NPE (RMSD 1.47 Å) using Swiss model (*Waterhouse et al., 2018*). For the active kinase domain model, we used a structure with the A-loop in DFG-in conformation that did not have an inhibitor bound to it (PDB-ID: 2G2I) and filled in missing P-loop residues using Modeller.

### Equilibrium MD

We used Gromacs (*Van Der Spoel et al., 2005*) version 2018.5 and 2020.3 along with the CHARMM36 force field (March 2019 version) (*Best et al., 2012*; *Huang and MacKerell, 2013*; *Klauda et al., 2010*). The proteins were placed in a dodecahedral simulation box with a distance of 2 nm between the box boundary and nearest protein atom. The box was filled with TIP3P water and 150 mM NaCl to neutralize charges. For energy minimization, we applied the steepest descent algorithm with a step size of 0.1 nm and a force tolerance of 1000 kJ/mol. During the two equilibration steps, the positions of the peptide backbone heavy atoms were restrained with a force constant of 1000 kJ/mol/nm. The temperature was equilibrated to 300 K using the v-rescale thermostat (*Bussi et al., 2007*) with a

coupling constant of 0.5 ps in a 100 ps simulation in the NVT ensemble. This was followed by a 1 ns simulation in the NPT ensemble to adjust the pressure to 1 bar using isotropic pressure coupling with the Berendsen barostat (*Berendsen et al., 1984*) with a coupling constant of 5 ps. After equilibration, we switched to the Parrinello–Rahman barostat (*Parrinello and Rahman, 1981*) for pressure coupling. For temperature coupling, we kept the v-rescale thermostat or switched to the Nosè–Hoover thermostat (*Nosé, 1984*; *Hoover, 1985*). We simulated 10 replicates for 2000 ns for each Abl model and thermostat, excluding the first 1000 ns for analysis (*Figure 2—figure supplement 3*). Since the sets of simulations with different thermostats show the same trends (*Figure 2—figure supplement 4*), we decided to combine their analyses. For integrating the equations of motion, the leap-frog integrator with a time step of 2 fs was used. All bonds involving hydrogen atoms were constrained using the LINCS algorithm (*Hess et al., 1997*). Van der Waals interactions were smoothly switched to zero between 1.0 and 1.2 nm using the force-switch method. Long-range Coulomb interactions beyond 1.2 nm were treated with the PME method (*Darden et al., 1993*).

## Force distribution analysis

For FDA, we concatenated the replicates of each model into a single trajectory and calculated the residue-based pairwise forces (*Costescu and Gräter, 2013*). We included all bonded and nonbonded interaction types and averaged the forces over the whole trajectory. We determined the force differences between (a) simulations with and without Myr using both kinase only (kinked $\alpha_I$ helix conformation) and SH3-SH2-kinase models and (b) simulations of kinase domain only versus SH3-SH2-kinase both in the absence and presence of Myr. We visualized force difference clusters with a minimum size of three residue pairs and a force threshold ranging between 20 and 100 pN using the fda_graph tool and VMD (*Humphrey et al., 1996*).

## Metadynamics

We used Gromacs 2018 patched with Plumed version 2.5.2 (*Bonomi et al., 2009*; *Tribello et al., 2014*) to perform the Metadynamics simulations. As the collective variable (CV) we chose the so-called *alpharmsd*, which compares backbone distances between adjacent amino acids to typical helix distances and thereby functions as a measure for helicity (*Pietrucci and Laio, 2009*). We performed the simulations on our SH3-SH2-kinase complex model without Myr and applied the bias to residues 515–521, which correspond to the unfolded region in the kinked $\alpha_I$ helix. We tested bias factors between 10 and 100, of which 20–30 gave the best results. Gaussians with an initial height of 1.2 kJ/mol and sigma of 0.02 and 0.03 were added every 500 integration time steps. The values of the CV and the bias potential were printed every 100 steps. Transient straightening events occurred in 11 out of 14 simulations. After straightening, the helix often partially unfolded, which is a consequence of Metadynamics penalizing the already visited conformations. We extracted a total of 23 frames with high alpharmsd from each Metadynamics run and simulated them for 500 ns as described above in equilibrium conditions.

## Umbrella sampling

We built a pure POPC membrane and embedded a myristoylated Gly residue such that the fatty acid chain is parallel to the membrane lipids and Gly faces the lipid head groups using Charmm-Gui (*Jo et al., 2008*; *Lee et al., 2016*). We considered SH3-SH2-kinase Abl models with a kinked, truncated, straight, or partially unfolded $\alpha_I$ helix conformation. For the truncated $\alpha_I$ helix, we deleted residues 518–531. Models with a straight or unfolded $\alpha_I$ helix were obtained from the Metadynamics simulations. We used frames from the end of the 500 ns equilibrium simulations as start frames for pulling simulations. Pulling and umbrella sampling was done similarly as reported previously (*Lemkul and Bevan, 2010*). We created a triclinic box with at least 2 nm distance between the protein and the box boundaries and elongated the box to 12 nm to provided space for pulling out Myr. The dimension for the membrane system was approximately 5 ×5 nm along the membrane plane and extended to 11 nm in the pulling direction vertical to the membrane plane. We used Gromacs 2020 for the umbrella sampling simulations. The systems were solvated, neutralized, energy minimized, and equilibrated as described in the 'Equilibrium MD' section. We pulled on the heavy atoms of the C-terminal NME cap of the Myr-Gly residue at 0.1 and 0.05 m/s with a spring constant of 500 kJ/mol/nm$^2$. Per model and pulling velocity, we performed at least three replicates.

Reference groups for pulling were the lipids of the upper membrane leaflet within a 1.5 nm radius around Myr, or the protein backbone residues of the four $\alpha$ helices comprising Abl's Myr binding site.

To obtain the starting configurations for the umbrella sampling windows, we extracted frames from the pulling trajectories with a spacing along the COM distance between pulling groups of 0.1 nm for the first 1 nm and 0.2 nm beyond that. This allows us to sample the initial unbinding stage in more detail. The fully unbound state is sufficiently covered at a distance of approximately 4.5 nm as evidenced by the plateau in the PMF plots. We simulated each window for 200 ns. For analysis with the weighted histogram analysis method (WHAM) (*Kumar et al., 1992*), we discarded the first 80 ns simulation time of each window (*Figure 5—figure supplement 1*). We estimated the error using bootstrapping with the b-hist method included in gmx wham (*Hub et al., 2010*) with 100 bootstraps.

## Acknowledgements

We acknowledge funding through the Deutsche Forschungsgemeinschaft (DFG, German Research Foundation) under Germany's Excellence Strategy – 2082/1-390761711, the Klaus Tschira Foundation, the state of Baden-Württemberg through bwHPC, as well as the DFG through grant INST 35/1134-1 FUGG. SdB thanks the Carl Zeiss Foundation for financial support.

## Additional information

### Funding

| Funder | Grant reference number | Author |
|---|---|---|
| Deutsche Forschungsgemeinschaft | 2082/1 - 390761711 | Svenja de Buhr<br>Frauke Gräter |
| Klaus Tschira Foundation | | Svenja de Buhr<br>Frauke Gräter |
| bwHPC | | Svenja de Buhr<br>Frauke Gräter |
| Deutsche Forschungsgemeinschaft | INST 35/1134-1 FUGG | Svenja de Buhr<br>Frauke Gräter |
| Carl Zeiss Foundation | | Svenja de Buhr |

The funders had no role in study design, data collection and interpretation, or the decision to submit the work for publication.

### Author contributions

Svenja de Buhr, Conceptualization, Data curation, Formal analysis, Visualization, Writing - original draft, Writing - review and editing; Frauke Gräter, Conceptualization, Formal analysis, Supervision, Funding acquisition, Writing - original draft, Project administration, Writing - review and editing

### Author ORCIDs

Svenja de Buhr ⓘ http://orcid.org/0000-0002-5368-3816
Frauke Gräter ⓘ http://orcid.org/0000-0003-2891-3381

### Decision letter and Author response

Decision letter https://doi.org/10.7554/eLife.85216.sa1
Author response https://doi.org/10.7554/eLife.85216.sa2

## Additional files

### Supplementary files
• MDAR checklist

## Data availability

Source data for all figures has been deposited on the Dryad Digital Repository under the DOI: https://doi.org/10.5061/dryad.9cnp5hqnx.

The following dataset was generated:

| Author(s) | Year | Dataset title | Dataset URL | Database and Identifier |
|---|---|---|---|---|
| de Buhr S, Gräter F | 2023 | Data from: Myristoyl's dual role in allosterically regulating and localizing Abl kinase | https://doi.org/10.5061/dryad.9cnp5hqnx | Dryad Digital Repository, 10.5061/dryad.9cnp5hqnx |

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
