## [Editor Report]

This is an important study of the mechanism of how binding of the fatty acid myristic acid (MYR) inhibits the activity of the kinase c-Abl, a critical regulator of many cellular processes. While the general aspects of this regulation are known from structure determination and biochemical studies, the exact molecular mechanism and the nature of the allosteric inhibition were not known. The authors use MD simulation to close this gap and provide a compelling mechanistic description of the inhibitory mechanism.

---

## [Decision Letter]

**Decision letter after peer review:**

Thank you for submitting your article "Myristoyl's dual role in allosterically regulating and localizing Abl kinase" for consideration by *eLife*. Your article has been reviewed by 2 peer reviewers, and the evaluation has been overseen by a Reviewing Editor and Volker Dötsch as the Senior Editor. The following individual involved in the review of your submission has agreed to reveal their identity: Roberto Covino (Reviewer #2).

The reviewers have discussed their reviews with one another, and the Reviewing Editor has drafted this to help you prepare a revised submission. As you will see, there is general support for the data presented, but some questions remain.

Essential revisions:

1) The overall simulation times seem to be rather short (several repeats, but only 500 ns) for such a large system with large conformational changes. There might be statistical convergence issues, especially because at least some of the starting structures were generated from available experimental structures after some modifications/modelling, and they might thus be out of equilibrium and need some time to fully relax during the MD simulations.

2) There do not seem to be convergence tests concerning the length of the simulations, which are usually considered to be standard analyses and a requirement for publication (Appendix Figure 5 shows the effect of different thermostats and capping of the peptide chain, but no tests concerning simulation time). This could be critical in the present case, where the authors acknowledge themselves (e.g., on p. 4) that there are only subtle differences between the different simulation systems and the variations within a given system are larger than the relevant (putative) differences between systems (Figure 1 C, D, E).

3) Issues with statistical convergence are expected not only for the standard MD simulations but also for the umbrella sampling simulations, as 50 ns sampling per window is nowadays not considered state of the art and is likely insufficient for quantitative binding free energy calculation, especially for membranes (see, e.g., DOI 10.1021/ct200316w).

4) Concerning the metadynamics simulations, these are usually done to obtain a free energy landscape. Why was this not attempted here? In the present case, the authors seemed to have used metadynamics only for generating starting structures, with different degrees of helicity of the α_I part, for subsequent standard MD simulations.

5) It would be superb if the authors could propose precise predictions that could inspire future experiments. Now that they present a residue-resolution allosteric pathway, can they suggest point mutations that would interrupt it? In addition, the evolutionary conservation of the residues identified to constitute the allosteric networks should be analysed.

6) The almost total absence of structural renders is surprising. Given the thorough discussion of structural details in the introduction, some renders would surely aid the reader.

*Reviewer #2 (Recommendations for the authors):*

– I was surprised about the almost total absence of structural renders. Given the thorough discussion of structural details in the introduction, some renders would surely aid the reader. I must admit that initially, I got a bit lost. Structural renders would help a lot also to appreciate the authors' mechanistic hypotheses.

– line 88 all systems contained ATP – this comes out of the blue, some explanation could be useful.

– Figure 1B. Please explain in the caption the inset and the color code. In general, the caption could give a more detailed explanation of all the symbols and colors used.

– Figure 5. Please make clearer in the caption what the long loop connecting Myr to the protein complex is.

– Regarding the free energy calculations, these are of course the most challenging ones from a technical point of view. The risk of these calculations is that there might be some hysteresis, that would impact the quantitative accuracy of the Δ Gs. Ideally, the protocol developed by Domanski and Best could help make these results more stable.

---

## [Author Response]

Essential revisions:Reviewer #2 (Recommendations for the authors):– I was surprised about the almost total absence of structural renders. Given the thorough discussion of structural details in the introduction, some renders would surely aid the reader. I must admit that initially, I got a bit lost. Structural renders would help a lot also to appreciate the authors' mechanistic hypotheses.

We added a new Figure 1 to the introduction comprising all mentioned structural motifs.

– line 88 all systems contained ATP – this comes out of the blue, some explanation could be useful.

We have clarified that ATP was included because it is the natural ligand for the kinase active site. Abl kinase has been suggested to bind ATP under physiological conditions, even in the inactive state. (Levinson, 2006, 10.1371/journal.pbio.0040144)

– Figure 1B. Please explain in the caption the inset and the color code. In general, the caption could give a more detailed explanation of all the symbols and colors used.– Figure 5. Please make clearer in the caption what the long loop connecting Myr to the protein complex is.

We aimed to make the captions more accurate and added additional information as suggested.

– Regarding the free energy calculations, these are of course the most challenging ones from a technical point of view. The risk of these calculations is that there might be some hysteresis, that would impact the quantitative accuracy of the Δ Gs. Ideally, the protocol developed by Domanski and Best could help make these results more stable.

Although we have considered the use of more elaborate methods such as coupling Umbrella Sampling with Replica Exchange as suggested by the reviewer, we reasoned that our simple Umbrella Sampling simulations are sufficient since we simulate the unbinding instead of the binding pathway. The latter comes with a much higher risk for hysteresis due to hitting potential unphysiological binding poses with distinct local minima. Also addressing the concerns of reviewer #1, we have significantly extended the simulation time invested into the free energy calculations, and now also added a new Figure 5 – figure supplement 1 to provide evidence for convergence of these simulations.